# Oral Immunization with Attenuated *Salmonella* Choleraesuis Expressing the P42 and P97 Antigens Protects Mice against *Mycoplasma hyopneumoniae* Challenge

Guodong Zhou,[a,b] Yichen Tian,[a,b] Jiashuo Tian,[a,b] Qifeng Ma,[a,b] Shan Huang,[a,b] Quan Li,[a,b] Shifeng Wang,[c] ⬤Huoying Shi[a,b,d]

[a]College of Veterinary Medicine, Yangzhou University, Yangzhou, Jiangsu, China
[b]Jiangsu Co-innovation Center for the Prevention and Control of Important Animal Infectious Diseases and Zoonoses, Yangzhou, China
[c]Department of Infectious Diseases and Immunology, College of Veterinary Medicine, University of Florida, Gainesville, Florida, USA
[d]Joint International Research Laboratory of Agriculture and Agri-Product Safety, Yangzhou University (JIRLAAPS), Yangzhou, China

**ABSTRACT** *Mycoplasma hyopneumoniae* (*M. hyopneumoniae*, Mhp) is the etiological agent of swine enzootic pneumonia (EP), which has been associated with considerable economic losses due to reduced daily weight gain and feed efficiency. Adhesion to the cilia is important for Mhp to colonize the respiratory epithelium. Therefore, a successful vaccine must induce broad Mhp-specific immune responses at the mucosal surface. Recombinant attenuated *Salmonella* strains are believed to act as powerful live vaccine vectors that are able to elicit mucosal immune responses against various pathogens. To develop efficacious and inexpensive vaccines against Mhp, the immune responses and protection induced by recombinant attenuated *Salmonella* vaccines based on the P42 and P97 antigens of Mhp were evaluated. In general, the oral inoculation of recombinant rSC0016(pS-P42) or rSC0016(pS-P97) resulted in strong mucosal immunity, cell-mediated immunity, and humoral immunity, which was a mixed Th1/Th2-type response. In addition, the levels of specific IL-4 and IFN-$\gamma$ in the immunized mice were increased, and the proliferation of lymphocytes was also enhanced, confirming the production of a good cellular immune response. Finally, both vaccine candidate strains were able to improve the weight loss of mice after a challenge and reduce clinical symptoms, lung pathological damage, and the inflammatory cell infiltration. These results suggest that the delivery of protective antigens with recombinant attenuated *Salmonella* vectors may be an effective means by which to combat Mhp infection.

**IMPORTANCE** Mhp is the main pathogen of porcine enzootic pneumonia, a highly infectious and economically significant respiratory disease that affects pigs of all ages. As the target tissue of Mhp infections are the mucosal sites of the respiratory tract, the induction of protective immunity at the mucosal tissues is the most efficient strategy by which to block disease transmission. Because the stimulation of mucosal immune responses is efficient, *Salmonella*-vector oral vaccines are expected to be especially useful against mucosal-invading pathogens. In this study, we expressed the immunogenic proteins of P42 and P97 with the attenuated *Salmonella* Choleraesuis vector rSC0016, thereby generating a low-cost and more effective vaccine candidate against Mhp by inducing significant mucosal, humoral and cellular immunity. Furthermore, rSC0016(pS-P42) effectively prevents Mhp-induced weight loss and the pulmonary inflammation of mice. Because of the effectiveness of rSC0016(pS-P42) against Mhp infection in mice, this novel vaccine candidate strain shows great potential for its use in the pig breeding industry.

**KEYWORDS** *Salmonella* Choleraesuis, *Mycoplasma hyopneumoniae*, P42, P97, recombinant vaccine

Address correspondence to Huoying Shi, hyshi@yzu.edu.cn.

The authors declare no conflict of interest.

Mycoplasma hyopneumoniae (*M. hyopneumoniae*, Mhp) is the main cause of *Mycoplasma* pneumonia of swine (MPS), which is also referred to as enzootic pneumonia (EP) (1). This disease is characterized by chronic nonproductive coughing, a poor growth rate and a poor feed conversion ratio (1, 2). Infected pigs often develop secondary infections, which makes Mhp a primary pathogen in the development of a respiratory disease complex in pigs and a major threat to the worldwide pig industry (3, 4).

At present, the treatment of this disease is mainly based on antibiotics and on the improvement of housing conditions and herd management practices, and while it does have some effectiveness in alleviating symptoms, it does not stop disease prevalence (5). Vaccination is the most cost-effective method by which to prevent Mhp infections in swine herds, and commercial vaccines include live attenuated vaccines and inactivated vaccines (6). In most cases, the prophylactic injection of the Mhp vaccine can decrease infection and can improve respiratory signs and growth performance (6–8). It is generally accepted that vaccine-induced antibody levels do not appear to be useful in assessing the severity of lung lesions in pigs, whereas mucosal and cellular immune responses are believed to play an important role in protection against Mhp (5, 9, 10). It might well be that the secretory immunoglobulin A (sIgA) prevents pathogens from adhering to the respiratory tract while IFN-$\gamma$ production enhances phagocytosis by alveolar macrophages (11, 12). However, inactivated vaccines have certain restrictions related to the method of presentation, resulting in a limited mucosa immune response, which requires adjuvants or immunostimulants to enhance the response (13). The 168 live vaccine strain (Mhp-168) is one of the Mhp vaccine strains that is used widely in China. However, immunization with the vaccine Mhp-168 induces a significant increase in IL-10 and a concurrent decrease in IFN-$\gamma$ expression, suggesting that vaccine Mhp-168 alone may lead to immune failure in practical applications (14). At the same time, Mhp has high requirements on the medium, and its high cost also limits the large-scale preparation and industrial production of whole bacterial vaccines (15). Therefore, there is an urgent need for a new type of vaccine that can induce durable mucosal, humoral, and cellular immune responses against Mhp in pigs at a lower cost and can achieve complete protection.

The identification of immunoprotective antigens is a crucial step in the development of effective vaccines against Mhp infection. In recent years, studies on the antigenic properties of Mhp revealed several immunodominant proteins, including the P36 cytoplasmic protein, the P65 and P74 cell membrane proteins, and the P97 adhesin (5, 6). These proteins are widely used to develop subunit vaccines and DNA vaccines (10, 16). The adhesion of Mhp to host tissues is the first step in successful colonization and infection. The C-terminal region of the adhesin P97 (R1 and R2), which is important among the different strains of Mhp, plays a significant role in the adherence (17). In pigs, infection triggers a strong immune response to P97 antigens in particular, which may prevent the adhesion of pathogens to epithelial cells (18). Consequently, the P97 adhesin protein has been the major target for vaccine development. Furthermore, antibodies specifically targeting the heat shock protein P42 of the Mhp effectively prohibit pathogen growth, making it a strong candidate for further development (19, 20). However, suitable methods of delivery are required for protein antigens, as they can be degraded due to their instability or can be taken up by inappropriate cells, and proteins presented in the absence of extraneous adjuvants are often weakly immunogenic.

*Salmonella* is a facultative intracellular bacterium that invades and persists inside host lymphoid tissues through the mucosa and continuously induces a strong mucosal and cellular immune response in the host (21, 22). A more effective and economical means by which to fight against many diseases is the live attenuated *Salmonella*-vectored vaccine, which could carry the antigen-coding genes of a pathogen and could induce humoral and/or cellular immune responses *in vivo* (23, 24). In addition, the use of *Salmonella* as a vector allows the placement of heterologous antigens directly into the lymphoid tissues, thereby permitting the maximum stimulation of the relevant

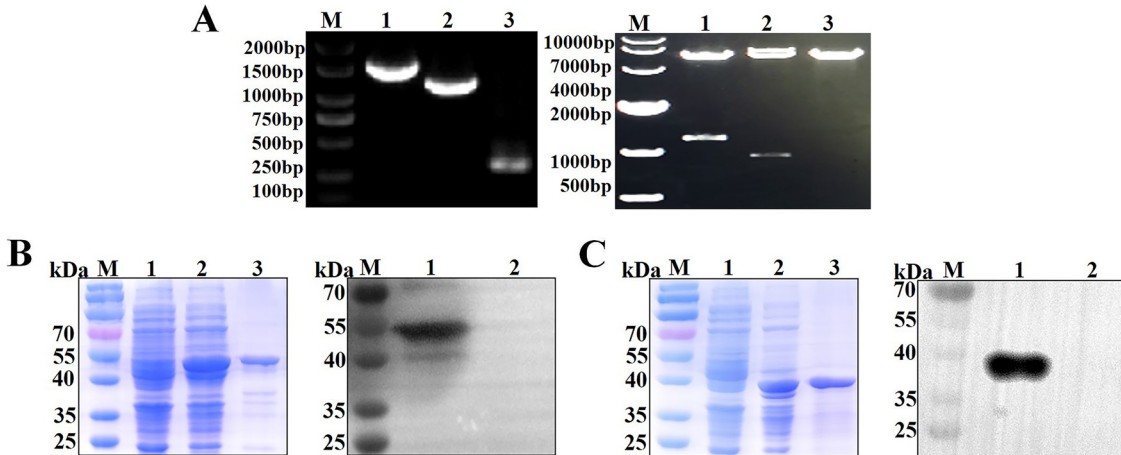

**FIG 1** Construction and purification of recombinant proteins. (A) Identification of recombinant plasmids pET-28a, pET-28a-P42, and pET-28a-P97 by PCR and restriction enzyme digestion. M, DNA marker (2,000 bp or 10,000 bp); lane 1, pET-28a-P42; lane 2, pET-28a-P97; lane 3, pET-28a. (B) Analysis of the purified P42 protein by SDS-PAGE and WB. SDS-PAGE: lane 1, BL21(pET-28a); lane 2, BL21(pET-28a-P42); lane 3, purified P42 protein. WB: lane 1, purified P42 protein; lane 2, BL21(pET-28a). (C) Analysis of the purified P97 protein by SDS-PAGE and WB. SDS-PAGE: lane 1, BL21(pET-28a); lane 2, BL21(pET-28a-P97); lane 3, purified P97 protein; WB: lane 1, purified P97 protein; lane 2, BL21(pET-28a).

lymphoid cells for the formation of sIgA (25, 26). This increases the possibility of developing vaccines that can elicit significant immune responses at mucosal sites beyond those achievable by other means (27). In a previous study, the delivery of the NrdF antigen of Mhp with an attenuated *Salmonella* vector (Δ*aroA*) induced significant cellular immune responses and sIgA responses and ameliorated lung pathological damage and daily weight gain in pigs, indicating that the delivery of the Mhp protective antigen with attenuated *Salmonella* is a feasible approach (28). However, with the *aroA* gene knockdown, the defects in the cell wall and outer membrane integrity of *Salmonella* may have a certain impact on its immunogenicity (29). In order to address these questions, Roy Curtiss and colleagues have described a variety of novel approaches for vaccine development, including regulated delayed attenuation, regulated delayed antigen synthesis, and regulated delayed lysis (for a balance between safety and immunogenicity) (30). At the same time, by constructing a balanced lethal system, the stability of the plasmid and the synthesis of foreign antigens are ensured (31). Based on these advantages, live attenuated *Salmonella* have been widely used as vaccine vectors for heterologous antigens that have been demonstrated to induce protective immune responses (32–35).

In previous studies, our laboratory constructed a new recombinant *Salmonella* Choleraesuis (*S.* Choleraesuis) attenuated vector rSC0016, which has a regulated delayed attenuation system and a delayed antigen expression system (32, 35). At the same time, the *sopB* gene was also knocked out to reduce the host intestinal inflammatory response caused by the *Salmonella* vector itself (34). The purposes of the present study were to construct recombinant *S.* Choleraesuis which expresses the P42 or P97 antigen of Mhp, and evaluate its potential immune responses in orally immunized mice. Meanwhile, a challenge experiment was carried out to determine the protective effect of the recombinant *Salmonella* vaccine after oral vaccination in a mouse infection model. This has provided a basis for research into the development of a vaccine against Mhp.

## RESULTS

**Recombinant protein expression and polyclonal antibody preparation.** The genes of Mhp F66 heat shock protein P42 and adhesin P97 were amplified via polymerase chain reaction (PCR), and then expressed in *E. coli* BL21(DE3), using the pET-28a vector. The success of the construction of the pET-28a-P42 and pET-28a-P97 plasmids were verified via PCR and enzyme digestion (Fig. 1A). Protein expression was then

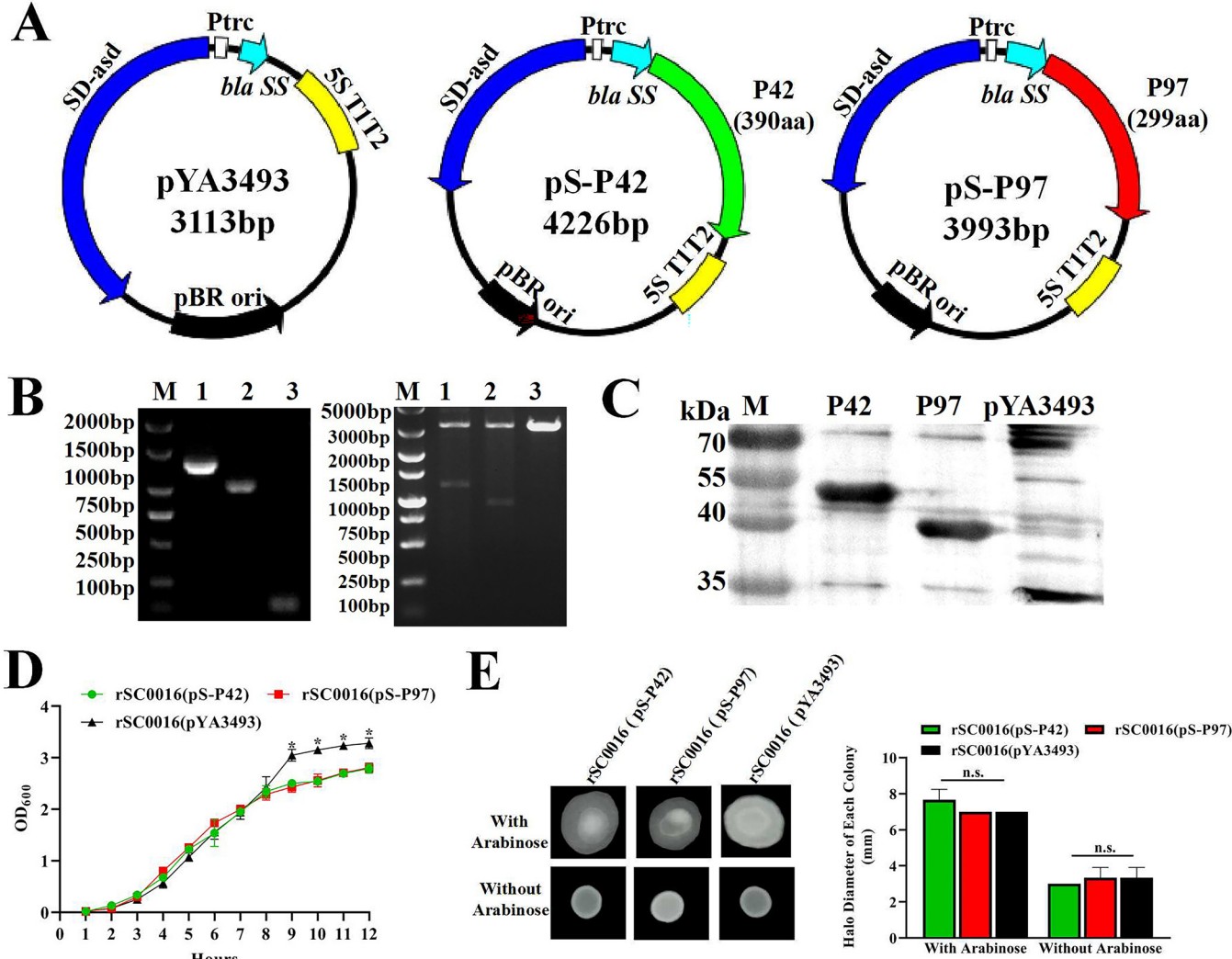

**FIG 2** Construction and phenotypic characterization of vaccine candidate strains. (A) Plasmid maps of vectors pYA3493, pS-P42, and pS-P97. (B) Identification of recombinant plasmids pYA3493, pS-P42, and pS-P97 by restriction enzyme digestion and PCR. M, DNA marker (2,000 bp or 5,000 bp); lane 1, pS-P42; lane 2, pS-P97; lane 3: pYA3493. (C) The expression of P42 and P97 in vaccine candidates was analyzed via WB. M, molecular weight markers; lane 2, recombinant protein P42; lane 3, recombinant protein P97; lane 4, empty vector plasmid (pYA3493). (D) Growth characteristics of rSC0016(pS-P42), rSC0016(pS-P97), and rSC0016(pYA3493) using the $OD_{600}$ values. (E) Motility was evaluated via the measurement of the growth halo of rSC0016(pS-P42), rSC0016(pS-P97), and rSC0016(pYA3493). Each experimental condition was conducted in duplicate, and three independent assays were performed in each case. All data are presented as means $\pm$ standard deviations (SD); n.s., nonsignificant ($P \geq 0.05$).

induced with 0.5 mM isopropyl-$\beta$-D-thiogalactoside (IPTG) at 37°C for 3 h with continuous shaking at 220 rpm. All proteins were expressed in soluble form and were purified by Ni-NTA resin. The purities of the P42 (Fig. 1B) and P97 (Fig. 1C) proteins were evaluated via SDS-PAGE and WB. Finally, the purified recombinant proteins were used for the production of polyclonal antibodies in mice.

**Construction and characterization of the live attenuated *S. Choleraesuis* vaccine strains rSC0016(pS-P42) and rSC0016(pS-P97).** Attenuated *Salmonella* has been shown to be a safe and effective vaccine carrier in previous studies and can induce high levels of mucosal, humoral, and cellular immune responses (36). The genes of Mhp heat shock protein P42 and adhesin P97 were amplified by PCR, using the F66 strain, and were cloned into the pYA3493 (Asd +) complemented plasmid (Fig. 2A). The insertion of the *P42* or *P97* gene cassette was confirmed by PCR amplification and by the double digestion of the host plasmid with the EcoRI and sall restriction enzymes (Fig. 2B). Finally, the constructed plasmid was transferred into the attenuated

*S.* Choleraesuis vector rSC0016 via electroporation to generate strains rSC0016 (pYA3493), rSC0016(pS-P42) and rSC0016(pS-P97). The expression of P42 and P97 in strain rSC0016 was evaluated by WB (Fig. 2C). Strains carrying the empty vector plasmids failed to produce P42-specific or P97-specific bands after reaction with the polyclonal antibodies. In addition, after the vaccine candidate strains were serially passaged in LB (about 50 passages), PCR and double-enzyme digestion were performed to verify that the results were the same as the initial passage strains, which proved that the plasmids carrying foreign antigens could exist stably in rSC0016 (data not shown). Growth curves of the rSC0016(pYA3493), rSC0016(pS-P42), and rSC0016(pS-P97) strains were compared using LB medium to determine the effects of heterologous antigens on bacterial growth. The results showed that the growth rate of rSC0016(pYA3493) was significantly higher than those of the rSC0016(pS-P42) and rSC0016(pS-P97) vaccine candidate strains at 9 h to 12 h, indicating that the presence of exogenous antigens had an effect on its growth characteristics (Fig. 2D). The motility assays of the vaccine candidate strains were determined in LB broth with or without 0.2% arabinose. No differences were observed for the growth halo of the rSC0016(pYA3493), rSC0016(pS-P42), and rSC0016(pS-P97) (Fig. 2E).

**Oral immunization with recombinant vaccine strains induces elevated antigen-specific antibody responses.** The degree of the host response toward the foreign antigens P42 and P97 were assessed by measuring the antigen-specific antibody responses in blood sera and mucosal secretions. The efficient production of both IgG and sIgA responses indicates the effectiveness of the rSC0016-delivered antigen presentation. Mice that were orally immunized with strains expressing either P42 or P97 developed high IgG titers against P42 or P97, respectively. The anti-P42 and anti-P97 serum titers again increased after booster immunization (Fig. 3A and B). Thus, these results indicated that immunization significantly prompted the serum IgG titer of the immunized group, compared to the PBS control and rSC0016(pYA3493) vector control.

The serum immune responses to P42 or P97 were further examined by measuring the levels of the IgG isotype subclasses IgG1 and IgG2a. As a result, both of the IgG subtypes were detected in the sera of mice immunized with rSC0016(pS-P42) and rSC0016(pS-P97), and the level of IgG2a was higher than that of IgG1, indicating that the candidate vaccine strains can induce a mixed Th1/Th2 immune response, with Th1 being predominant (Fig. 3A and B).

To determine whether the mice orally immunized with recombinant vaccine strains developed mucosal immune responses, the level of specific sIgA antibodies in the mice were measured via iELISA in the BALs (Fig. 3C and D). As shown in Fig. 3C, at the 2-week time point, the BALs of the group with rSC0016(pS-P42) contained specific sIgA antibodies against the recombinant P42 protein. By 5 weeks, after booster immunization, the levels of P42-specific sIgA antibodies showed a significant increase. The P97-specific sIgA response of the rSC0016(pS-P97) immunized group showed a similar trend to the rSC0016(pS-P42) group (Fig. 3D), and after the booster immunization, those of the two immunization groups were significantly higher than those of the rSC0016(pYA3493) and PBS control groups.

As expected, immunization with rSC0016(pS-P42), rSC0016(pS-P97), or rSC0016 (pYA3493) significantly prompted the serum anti-OMPs titers of the immunized group, compared to the PBS control (Fig. 3E).

**Oral immunization of recombinant vaccine strains induces significant cellular immune responses and antigen-specific IFN-$\gamma$ and IL-4 production.** 1 week after the booster immunization, lymphocyte proliferation responses were detected by CCK8 in the immunized mice. As shown in Fig. 4A, P42-specific and P97-specific proliferation responses were induced in the lymphocytes of mice orally immunized with rSC0016 (pS-P42) or rSC0016(pS-P97), respectively. The proliferative response of lymphocytes stimulated by P42 was significantly higher in the rSC0016(pS-P42) group than in the PBS group or the empty vector control group. The SI value of the rSC0016(pS-P97) immunized group was higher than that of the PBS or the empty vector control group, but the difference was not statistically significant.

To further evaluate the type of immune response (Th1 or Th2), ELISPOT was used to

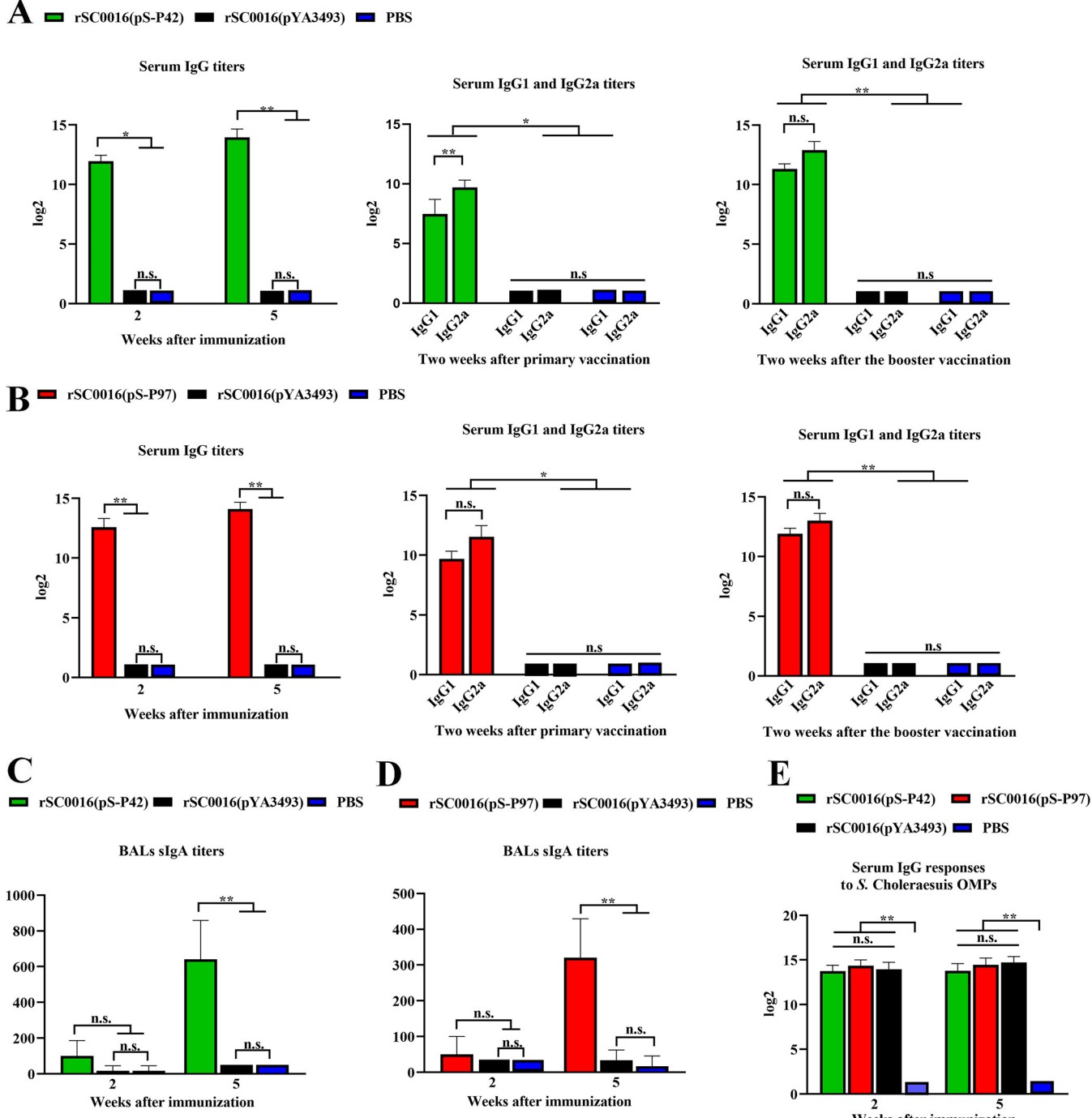

**FIG 3** Antibody responses induced by vaccine candidates in mice. P42-specific IgG antibodies and isotype responses (A). P97-specific IgG antibodies and isotype responses (B). The BALs sIgA responses to P42 (C) and P97 (D) and the serum IgG responses to *S. Choleraesuis* OMPs (E) were assayed by ELISA. BALs samples were collected at 2 and 5 weeks following the primary vaccination. Error bars represent the variation between different mice. The results are expressed as the mean ± SD. The degrees of statistical significance are indicated as follows: *, $P < 0.05$; **, $P < 0.01$; n.s., nonsignificant ($P \geq 0.05$).

compare the P42 or P97 stimulation of IFN-$\gamma$ (Th1-associated) and IL-4 (Th2-associated) production by spleen cells from immunized and control mice. As shown in Fig. 4B and C, the oral administration of rSC0016(pS-P42) or rSC0016(pS-P97) significantly upregulated the corresponding IFN-$\gamma$/IL-4 secreting cell amount, compared to those of the PBS and rSC0016(pYA3493) groups. Moreover, the number of cells secreting IFN-$\gamma$ was higher than that of IL-4, indicating that the recombinant candidate vaccine strains can induce a Th1-biased immune response. This finding is consistent with the detection

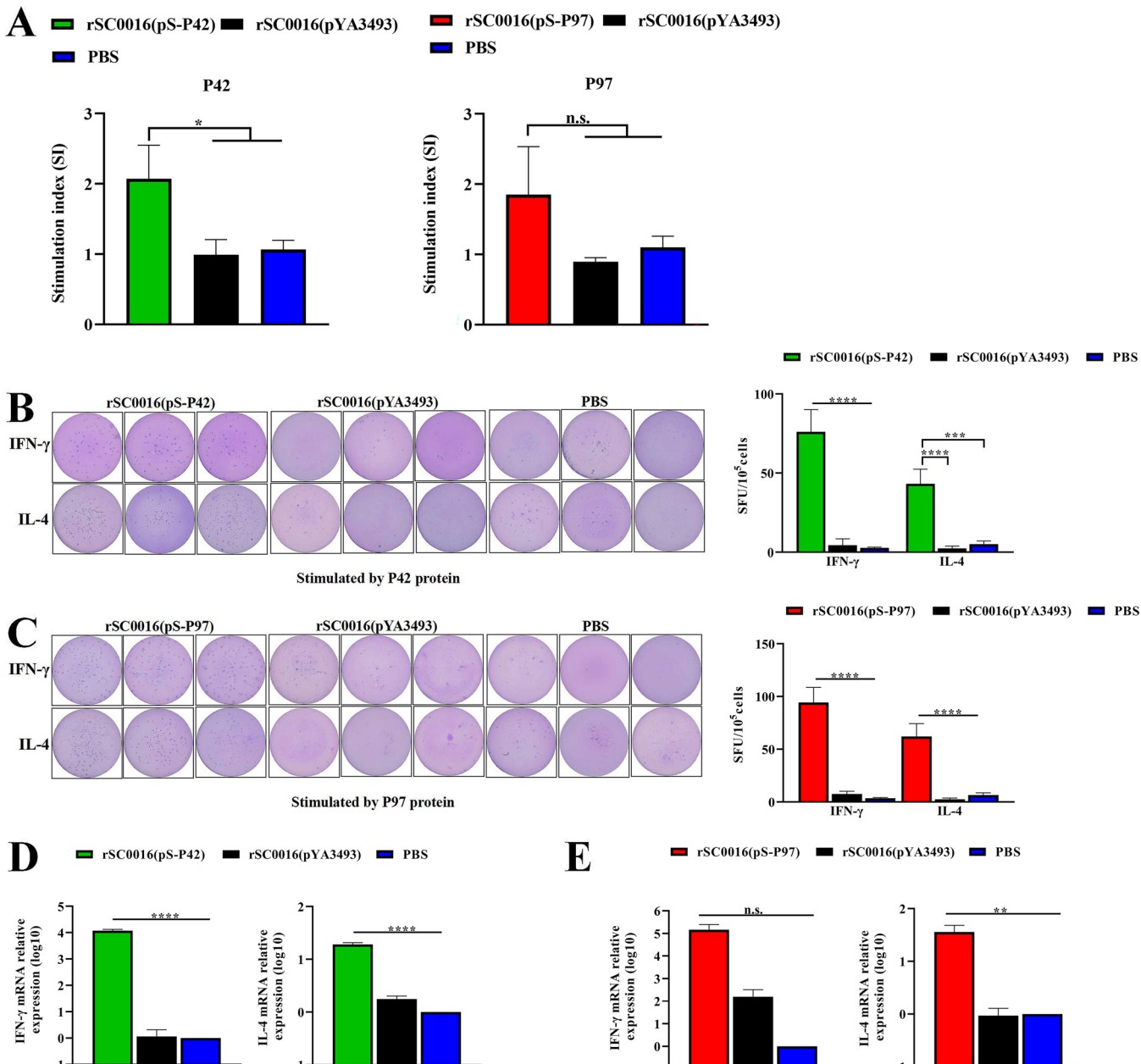

**FIG 4** Cell-mediated immune responses and antigen-specific stimulation of IFN-γ and IL-4 production. Splenic lymphocytes were performed on BALB/c mice sacrificed on day 7 following boost vaccination. PBS controls were also included. (A) Lymphocyte proliferation was measured by the CCK-8 method as described in the text and is shown as an SI. Splenic lymphocytes were adjusted to a density of $5 \times 10^5$ cells/mL and were plated into 96-well ELISpot plates coated with anti-mouse IFN-γ and IL-4 antibodies at 100 μL/well. The cells were restimulated for 48 h with 0.1 μg/well of recombinant P42 (B) or P97 (C) for ELISPOT assays. Lymphocytes were prepared and cultured with either the P42 antigen or the P97 antigen for 48 h. The mRNA expression of cytokines IFN-γ and IL-4 (D, E) were analyzed via qRT-PCR. The results are expressed as the mean ± SD. The degrees of statistical significance are indicated as follows: *, $P < 0.05$; **, $P < 0.01$; n.s., nonsignificant ($P \geq 0.05$).

results of IgG1 and IgG2a in serum. Finally, the cytokine mRNA levels in the splenocytes were analyzed by qPCR, and β-actin was used as the reference protein (Fig. 4D and E). As expected, the results of the mRNA expression were in accordance with the ELISPOT results, as indicated in Fig. 4B and C.

**The recombinant vaccine improves body weight changes and sneezing symptoms in mice.** The body weight change and severity of sneezes of each mouse were closely monitored. The results of the body weight changes are shown in Fig. 5A. There were no statistically significant differences between the groups in terms of body weight before the challenge. After the challenge, compared with the blank control

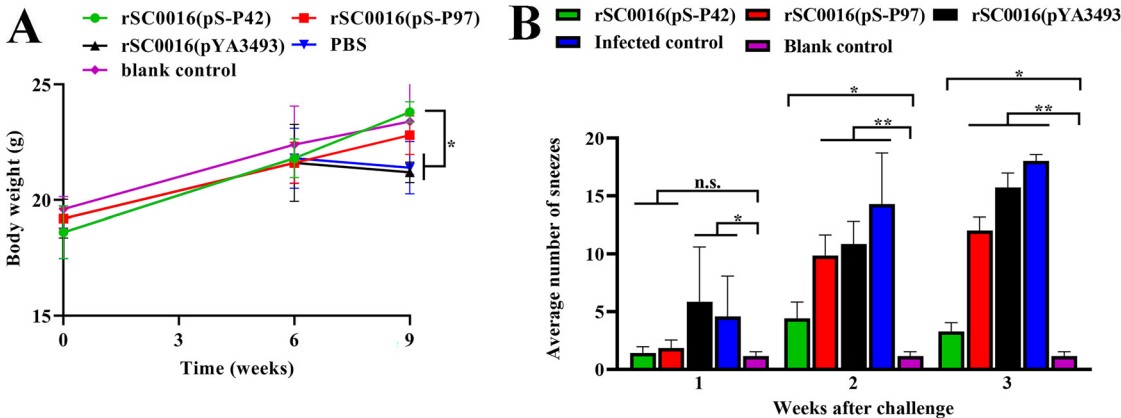

**FIG 5** Weight of mice and average number of sneezes. Before and after the challenge, changes in body weight (A) were monitored. The number of sneezes was also recorded. After the Mhp F66 challenge, each individual mouse was observed for 10 min a day, and the average number of sneezes per week was recorded (B). The results are expressed as the mean ± SD. The degrees of statistical significance are indicated as follows: *, $P < 0.05$; **, $P < 0.01$; n.s., nonsignificant ($P \geq 0.05$).

group, the body weights in the rSC0016(pYA3493) and PBS immunized groups were significantly decreased. In contrast, the body weights of the recombinant vaccine groups did not show significant changes, compared to the blank control group. Among them, the rSC0016(pS-P42) group had the most obvious effect in terms of the improvement of weight gain. At the same time, mice were continuously observed for 21 days after the challenge (10 min per day), and the number of sneezes of the mice in each group was scored. As shown in Fig. 5B, sneezing was significantly increased in the PBS and empty vector control groups, compared with the blank control group during the 21-day post-challenge observation period. The numbers of sneezes in the mice immunized with rSC0016(pS-P97) were lower than those in the mice immunized with PBS or rSC0016(pYA3493), but they were still significantly higher than those of the blank control group at weeks 2 and 3. Similar sneezing symptoms were also observed in the rSC0016(pS-P42) group following the challenge, but the increases were lower than those observed in the other immunization groups.

**Oral immunization with rSC0016(pS-P42) or rSC0016(pS-P97) ameliorates microscopic lung lesions in mice.** A histological examination revealed that there were obvious histopathological changes in the lung tissues of the mice in the PBS or rSC0016(pYA3493) group, including obvious inflammatory cell infiltration, thickened alveolar walls, alveolar congestion, and bleeding. Furthermore, compared with the PBS and rSC0016(pYA3493) immunized groups, the rSC0016(pS-P42) and rSC0016(pS-P97) immunized groups showed mild hyperplasia of the lymphoid tissue and less inflammatory cell infiltration (Fig. 6A). The H&E staining results showed that the alveolar morphology of the lung tissue from the blank control group was normal and that there was no inflammatory cell infiltration. In order to more directly evaluate the protective effect of the vaccine, the lung tissue lesions were scored (from 0 to 3) based on the severity of the peribronchiolar and perivascular lymphoid tissue hyperplasia (Fig. 6B). Among them, the rSC0016(pYA3493) and PBS immunization groups had the most serious pathological tissue damage and the highest scores, which significantly differed from those of the blank control group. Although the rSC0016(pS-P97) immunized group had a certain protective effect, its score was still significantly higher than that of the blank group. Finally, the rSC0016(pS-P42) immunized group displayed better protection against Mhp, and the lung pathological damage was milder after the challenge. The score of the rSC0016(pS-P42) immune group was similar to that of the blank control group, and there was no significant difference. The above-described experimental data show that immunizing mice with rSC0016(pS-P42) and rSC0016(pS-P97) can reduce pulmonary inflammation and can improve the lung injury score.

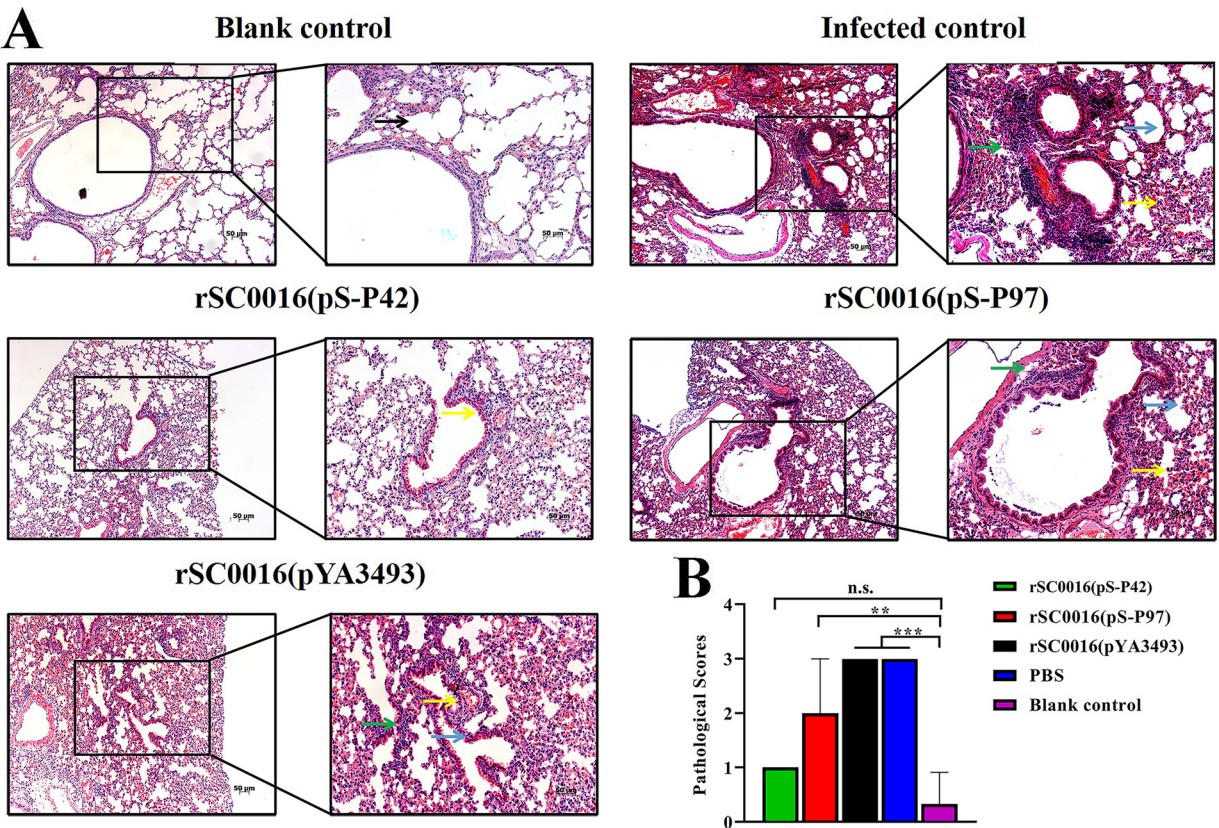

**FIG 6** Histological analysis and pathological scores in the lungs. H&E staining revealed the histological characteristics of the challenged mice (A) (magnification, ×100 and ×200; scale bar, 50 μM). The lesions in the infected control (PBS-vaccinated and challenged) and rSC0016 (pYA3493) groups were more extensive. There was a large amount of lymphocyte infiltration (green arrow), hemorrhage (yellow arrow) and alveolar interstitial wall thickening around the bronchus (blue arrow). The lesion range of the rSC0016(pS-P97) group was relatively limited. However, there were also many pathological changes, such as lymphocyte infiltration (green arrow), hemorrhage (yellow arrow), and alveolar interstitial wall thickening (blue arrow). The rSC0016(pS-P42) group showed only slight hemorrhage (yellow arrow) and essentially no obvious pathological changes, compared with the blank control (black arrow represents a normal alveolar structure). The histological characteristics of the mice in the different immune challenge groups were scored as described in the Materials and Methods section (C). The results are expressed as the mean ± SD. The degrees of statistical significance are indicated as follows: *, $P < 0.05$; **, $P < 0.01$; n.s., nonsignificant ($P \geq 0.05$).

## DISCUSSION

The target tissue of Mhp infection is the mucosal site of the respiratory tract, which can cause coughing, wheezing, a loss of appetite, and many other symptoms in infected herds (9). Therefore, an ideal preventive vaccine must simultaneously elicit mucosal immunity that prevents Mhp transmission and systemic immunity that controls lung pathological tissue damage. Live recombinant *Salmonella* vectors deliver heterologous antigens through the mucosal surfaces as a powerful strategy for inducing mucosal and systemic immune responses (36). Following oral vaccination, *Salmonella* invades the mucosal surfaces and spreads through the mesenteric lymph nodes to distant sites, such as the spleen and liver, which often results in the induction of mucosal, systemic cellular, and humoral immune responses (21, 22, 36). All of these suggest that the presentation of Mhp antigens by *Salmonella* is a strong candidate for vaccine development.

sIgA is the major contributor to mucosal immune function and plays a crucial role in the defense against pathogens (12). Several studies have shown that sIgA can bind to microorganisms and prevent them from attaching to or penetrating epithelial cells, such as *Streptococcus pneumoniae*, *Neisseria meningitidis*, and *Bordetella pertussis* (37–39). Furthermore, vaccine-induced mucosal immunity also appears to play an important role in the control of Mhp infection (40). The P97 protein is one of the earliest discovered and most intensively studied adhesin proteins in Mhp (17). Antibodies against the P97 protein have been shown to prevent

pathogens from adhering to airway epithelial cells (18). In this study, the specific sIgA against recombinant protein P97 could be detected in the BALs of mice immunized with rSC0016 (pS-P97), and it was found to be significantly increased after booster immunization. This is consistent with previous studies that show that intranasal immunization with rAdP97c resulted in mucosal and systemic immune responses (41). In addition to P97, the P42 protein is also used as one of the main immunogenic proteins for the development of genetically engineered vaccines (19, 20). A previous study indicated that mice that were intranasally immunized with a subunit vaccine containing P42 fused to the B subunit of the heat-labile enterotoxin of *Escherichia coli* induced the highest mucosal immune responses (42). This may be due to the fact that the P42 protein is a member of the heat shock protein family that functions as an immunostimulant (43). Moreover, in this study, rSC0016(pS-P42) also induced elevated P42-specific sIgA, which was significantly higher than that observed in the PBS or empty vector control group. Previous work has revealed that the production of IgG can enhance opsonophagocytosis against Mhp infections and can reduce the area of pneumonia lesions (19, 44, 45). In our results, the immunization of mice with rSC0016(pS-P42) and rSC0016(pS-P97) significantly induced specific anti-P42 and anti-P97 IgG. Enhanced phagocytosis and enhanced antigen presentation have been described for a variety of pathogens with opsonizing antibodies that may contribute to resistance to infection (46). These results indicate that both of the vaccine candidate strains induce potent mucosal and systemic humoral immune responses.

Furthermore, it was proposed that the cellular immune response plays an important role in enhancing the protective immunity against Mhp infections (40). CD4$^+$ T cells play a major role in coordinating immune responses (47). On the basis of the distinct populations of cytokine profiles, CD4$^+$ T cells can be classified into T helper (Th) 1 or 2 cells (48). Th1 cells are involved in cell-mediated immune responses and activate B cells to produce opsonizing antibodies, such as IgG2a, while Th2 cells contribute to humoral immunity, promoting the secretion of IgG1 and IgA (48, 49). Here, we have demonstrated that recombinant rSC0016(pS-P42) and rSC0016(pS-P97) were capable of inducing both Th1 and Th2 responses to P42 and P97, as evidenced by the production of both the IgG1 and IgG2a isotypes in the immunized groups. King et al. have reported that the immunization of pigs with recombinant adhesion P97 did not induce significant protection in challenge experiments (50). However, the results of Chen et al. showed that when the C-terminal region of P97 was fused to the *Pseudomonas aeruginosa* toxin A, both immunized mice and pigs could generate a specific immune response against P97 (51). Moreover, the delivery of the P97 protein with recombinant *Erysipelothrix rhusiopathiae* can reduce the lung tissue damage caused by Mhp infections in pigs, and mice can be immunized with replication-deficient adenovirus vectors to induce systemic Th1/Th2 immune responses and mucosal immune response (45, 52). These studies suggested that the different vectors, adjuvants, and routes of immunization of the vaccines could induce different types of immune responses. In previous studies, *Salmonella* as a vector preferentially promotes Th1-type immune responses to heterologous antigens (33, 35).

As previously mentioned, the oral administration of recombinant rSC0016(pS-P42) and rSC0016(pS-P97) could stimulate Th cell differentiation and could modulate the immune response (IgG1 or IgG2a). In addition to the role of T cells in the switching of IgG subtypes, the cytokines secreted by T cells are also critical in preventing pathogen infections (49). Th1 cytokines, such as IFN-$\gamma$, promote CD8$^+$ T cell responses and induce murine B cells to switch to an IgG2a isotype (53). In contrast, Th2 cytokines (such as IL-4 and IL-5) promote the class switching of B cells to neutralizing antibodies (such as IgG1), and they further modulate the magnitude of the Th1 cytokine responses (54, 55). After oral immunization, we observed that recombinant rSC0016(pS-P42) and rSC0016(pS-P97) could induce significant P42-specific or P97-specific IFN-$\gamma$ and IL-4 production, compared to the control groups. While the classical effector functions of CD8$^+$ T cells may not be associated with immune responses against *Mycoplasma* species that are not intracellular organisms, mouse models suggest that CD8$^+$ T cells

suppress the inflammatory responses mediated by CD4$^+$ T cells (56). Furthermore, CD4$^+$ T cells (mainly Th1 cells) are essential for disease prevention, possibly due to the activation of IFN-$\gamma$ receptors on macrophages (11). In addition, the P42 or P97 antigen could significantly stimulate the specific lymphocyte proliferation responses of immunized mice *in vitro*. All of these results demonstrated that recombinant rSC0016(pS-P42) or rSC0016(pS-P97) could induce cell-mediated immune responses.

Mhp infection mainly causes decreased daily weight gain and lung damage in pigs, and commercial vaccines have been shown to improve the feed conversion rate, increase the daily weight gain, and greatly reduce the clinical symptoms and lung damage (2, 8). In this study, mice in each immunization group were challenged with virulent strain F66. Compared with the blank control group, all of the mice in the empty vector group and in the PBS immunized group developed clinical symptoms, such as sneezing, chills, and weight loss. This is in accordance with most of the previous studies in pigs (1). However, after immunizing mice with the candidate vaccines rSC0016(pS-P42) and rSC0016(pS-P97), they can improve the clinical symptoms of the respiratory tract, reduce the number of sneezes, and alleviate the impact on the weights of the mice. The main symptom of Mhp in pigs is dry cough (1). In this study, dry cough was also observed in a small number of mice, but it was not statistically significant. In our study, most of the mice developed sneezing symptoms after the challenge. This may be related to the colonization of Mhp in the respiratory tract, and the destruction of the ciliary structure makes the mice more sensitive to stimuli (57). Similarly, lung lesions suggested that the rSC0016(pS-P42) and rSC0016(pS-P97) immunized groups showed mild histopathological changes and only a small amount of inflammatory cell infiltration in the lung. In contrast, the histopathological findings of the lungs in the empty vector and the PBS immunized group showed that the alveolar septum was significantly widened, peribronchiolar and perivascular lymphoid tissue hyperplasia, and inflammatory cell infiltration, which are consistent with previous experimental results in pigs (58). Overall, these results demonstrate that immunizing twice with the recombinant vaccine candidate strains induces protection against a Mhp challenge. However, it should be further explored whether the lesions of different lung lobes in mice are consistent after a Mhp challenge. In the infected pigs, the lung lesions presented a distinct border, and the severity of the lesions varied in the different lobes (44).

In conclusion, we have demonstrated that the oral live vaccine prepared in this study is capable of inducing specific local mucosal, humoral, and cellular immunities in mice and that *Salmonella* is a potential vaccine vector with which Mhp antigens can be delivered. Here, rSC0016(pS-P42) and rSC0016(pS-P97) were able to reduce clinical symptoms and microscopic lung lesions, with the rSC0016(pS-P42) delivering the heat shock protein P42 being the most effective. Therefore, the two recombinant vaccine strains have the potential to control swine enzootic pneumonia, and this needs to be further studied in pig populations.

## MATERIALS AND METHODS

**Plasmids, bacterial strains, and growth conditions.** The strains and plasmids used in this work are shown in Table 1. *Escherichia coli* (*E. coli*) DH5$\alpha$ competent cells were used for the cloning experiments. The bacterial strain *E. coli* $\chi$7213 and plasmid pYA3493 were kindly provided by Roy Curtiss III (The Biodesign Institute, Arizona State University, Tempe, AZ). The Mhp virulent strain F66 (CVCC354) and the *S.* Choleraesuis strain C78-3 (CVCC79103) were purchased from the China Institute of Veterinary Drugs Control (CVCC, Beijing, China). The balanced-lethal Asd$^+$ plasmid vectors pS-P42 and pS-P97 were constructed by cloning the DNA sequence of *P42* and *P97* (full-length *P42* gene and the C-terminal region [R1R2] of *P97*) into pYA3493. The Mhp strain F66 was cultured in modified Friis medium. The *E. coli* $\chi$7213 and *S.* Choleraesuis strains were grown in Luria-Bertani (LB) broth or on LB agar plates. When required, the medium was supplemented with 50 $\mu$g/mL ampicillin (Amp), 50 $\mu$g/mL chloramphenicol (Cm), 50 $\mu$g/mL kanamycin (Km), or 50 $\mu$g/mL 2,6-diaminopimelic acid (DAP). The growth media for the vaccine strains were supplemented with 0.2% arabinose and 0.2% mannose.

**Ethics statement.** Female BALB/c mice were purchased from the Comparative Medicine Center, Yangzhou University (Jiangsu, China). All of the animal experiments were performed in strict accordance with the animal welfare standards of the Animal Research Committee Guidelines of Jiangsu Province (License Number: SCXK(SU)2017-0007) and were approved by the Ethics Committee for Animal

**TABLE 1** Bacterial strains and plasmids used in this study

| Strains and plasmids | Characteristics[a] | Source, reference, or function |
|---|---|---|
| **Strains** | | |
| BL21 (DE3) | For expression of the recombinant plasmids | Invitrogen |
| $\chi$7213 | *Thi 1 thr 1 leuB6 fhuA21 lacY1 glnV44 asdA4 recA1 RP4 2 Tc::Mu pir* | Provided by Roy Curtiss III |
| C78-3 | Wild type, virulent, CVCC79103 | CVCC |
| rSC0016 | $\Delta P_{crp527}$::TT *araC* $P_{BAD}$ *crp* $\Delta pmi$-2426 $\Delta relA199$::*araC* $P_{BAD}$ *lacI* TT $\Delta sopB1686$ $\Delta asdA33$ | 35 |
| Mhp-F66 | Wild type, highly virulent strain, CVCC354 | CVCC |
| **Plasmids** | | |
| pYA3493 | Plasmid Asd +; pBR *ori*, $\beta$-lactamase signal sequence-based periplasmic secretion | 31 |
| pET28a | Expression vector, Kan[r] | Novagen |
| pMD19-T | Cloning vector; Amp[r] | TaKaRa |
| pS-P42 | pYA3493 with P42 | This study |
| pS-P97 | pYA3493 with P97 | This study |

[a]Kan[r], kanamycin resistance; Amp[r], ampicillin resistance.

Experimentation of Yangzhou University. For the animal experiments, all efforts were made to minimize suffering and to maximize animal welfare.

**Protein expression, protein purification, and antibody preparation.** To generate purified proteins, the target genes were amplified via PCR, digested by EcoRI/SalI (TaKaRa), and then inserted into the plasmid pET-28a (Novagen) to transform *E. coli* BL21(DE3) competent cells (Invitrogen) via calcium chloride transformation. Furthermore, the positive clone was induced by isopropyl-$\beta$-D-thiogalactoside (IPTG) (Sangon), and the target proteins were expressed and purified by nickel-nitrilotriacetic acid (Ni-NTA) resin. The concentrations of the purified proteins were detected via BCA methods (Thermo Fisher). Finally, the recombinant proteins were analyzed via sodium dodecyl sulfate-polyacrylamide gel electrophoresis (SDS-PAGE) and Western blotting (WB). The polyclonal antisera of P42 and P97 were prepared following our previous protocol with minor modifications (33). Briefly, 6-week-old mice (female) were injected intramuscularly in the hind legs two times. The first immunization contained 20 $\mu$g protein mixed with 50 $\mu$L Quick adjuvant (Biodragon). 3 weeks later, the mice were immunized once again. At 35 days post-boost, blood was abstracted from the tails of the mice and tested via indirect enzyme-linked immunosorbent assay (iELISA).

**Construction of vaccine strains and assessment of plasmid stability.** The attenuated *S.* Choleraesuis vector rSC0016 was previously constructed in our laboratory, and its *sopB* gene was identified via PCR to ensure the correctness of the vector (data not shown). The insert fragments were digested with EcoRI/SalI (TaKaRa) from the pMD19-T vector (TaKaRa) and were then ligated into the expression vector pYA3493 (Asd +) that had been digested with the same enzymes, resulting in the constructed vectors (pS-P42 or pS-P97). Finally, the recombinant plasmids were transformed into rSC0016 via electroporation as previously described, and the recombinant *S.* Choleraesuis strains were named rSC0016(pYA3493), rSC0016(pS-P42), and rSC0016(pS-P97) (35). In order to verify whether the P42 and P97 proteins could be expressed normally in the two vaccine candidate strains, WB was carried out with the above-prepared poly-antibody sera. The constructed vaccine strains were serially passaged, and, after culturing for 50 generations, PCR and the double digestion of the plasmid were used to verify whether the plasmid could exist stably.

**Growth characteristics and motility analysis of vaccine candidate strains.** Bacterial growth curves were measured by the method described previously (33). The bacterial cultures were incubated at 37°C, and the $OD_{600}$ value of the cultures was measured at 1 h intervals for 12 h. LB plates containing 0.5% agar with or without arabinose (Sigma) were inoculated with each strain for the motility assays (59). The assays were repeated three times.

**Immunization and challenge.** The vaccine strains used in this study were prepared using the method described in previous studies (33, 35). Female BALB/c mice (6 weeks old) were divided into five groups (20 mice/group) in the experiments. The groups were divided as follows: group 1, unvaccinated and unchallenged (blank control); group 2, PBS-vaccinated and challenged (infected control); group 3, rSC0016(pYA3493) vaccinated and challenged (empty vector control group); group 4 and 5, rSC0016(pS-P42) and rSC0016(pS-P97) orally (the mice were inoculated with 20 $\mu$L of PBS containing $1 \times 10^9$ CFU of each strain), respectively. Each animal was boosted with the same dose 21 days after the first inoculation. Sera were extracted from the blood via mandibular vein puncture at 2 and 5 weeks after the primary vaccination. Bronchoalveolar lavage fluids (BALs) samples from five mice per immunity group were simultaneously collected and stored at $-20$°C until used. The challenge was performed at day 21 after the boost vaccination with $10^8$ CCU of the F66 strain via intraperitoneal and nasal routes.

**Immune responses by indirect enzyme-linked immunosorbent assay.** The sera from all of the mice in a group were pooled for analysis. An indirect enzyme-linked immunosorbent assay was used to measure the IgG antibodies against the serovar Choleraesuis OMPs, recombinant proteins (P42 or P97) in serum, IgG1 and IgG2a in serum, and sIgA in BALs against recombinant proteins, as previously described (19, 33, 35). In brief, 100 $\mu$L solutions containing 0.2 $\mu$g/well of either *Salmonella* OMPs or recombinant proteins in sodium carbonate-bicarbonate coating buffer (pH 9.6) were used to coat 96-well ELISA microtiter plates. The plates were then incubated overnight at 4°C. After washing with PBST (PBS with 0.1% Tween 20), the plate was blocked with 5% skim milk powder in PBS for 2 h at 37°C. Then, 1:200 diluted serum or 1:10 diluted BALs suspended in PBS were added to the plate wells for binding.

**TABLE 2** The primer information

| Primer name | Sequences (5′–3′)[a] | References |
|---|---|---|
| *P42*-F | CGGAATTCCTTGTTAAAAAAATCAAAGAAGAA | This study |
| *P42*-R | GCGTCGACTTAATCCTGCTTGATTTCAG | |
| *P97*-F | CGGAATTCACTACAAAAGAAGGTAAAAGAGAA | This study |
| *P97*-R | GCGTCGACTTATTTAGATTTTAATTCCTGATT | |
| pYA3493-F | AACGCTGGTGAAAGTAAAAGATG | This study |
| pYA3493-R | CAGACCGCTTCTGCGTTCT | |
| pET-28a-F | TAATACGACTCACTATAGGG | This study |
| pET-28a-R | GCTAGTTATTGCTCAGCGG | |
| *IFN-γ*-F | TCAAGTGGCATAGATGTGGAAGAA | 60 |
| *IFN-γ*-R | TGGCTCTGCAGGATTTTCATG | |
| *IL-4*-F | ACAGGAGAAGGGACGCCAT | 60 |
| *IL-4*-R | GAAGCCCTACAGACGAGCTCA | |
| *β-actin*-F | CCCTAAGGCCAACCGTGAA | 60 |
| *β-actin*-R | CAGCCTGGATGGCTACGTACA | |

[a]Underlined nucleotides denote enzyme restriction sites.

After 2 h of incubation at 37°C and washing, the plate was further incubated with 100 $\mu$L of 1:5,000 diluted goat anti-mouse IgG (or IgG1 or IgG2a) antibodies and goat anti-mouse IgA antibodies (Sigma, St. Louis, MO, USA), all of which conjugated with HRP at 37°C for 1 h. After washing, TMB substrate solution (Solarbio, Beijing, China) was added for coloration for 15 min at 37°C, and the reaction was stopped by the addition of 2 M $H_2SO_4$. The sample wells displaying $OD_{450}$ values above 2.1 times those of the negative-control wells were judged as positive.

**Specific lymphocyte proliferation response assay.** The cell counting kit CCK-8 (Sangon) detection method was used. Spleens (five mice in each group) were harvested under sterile conditions 1 week after the final immunization, and the splenic lymphocytes were isolated by using mouse lymphocyte separation medium (Dakewe Biotech, China). Then, the suspensions were plated in triplicate on to 96-well plates at a concentration of $5 \times 10^5$ cells/well, and they were incubated at 37°C with 5% $CO_2$ for 24 h with the corresponding purified P42 and P97 antigens individually at final concentrations of 10 $\mu$g/mL. At 2 h before the end of the incubation period, 10 $\mu$L of CCK-8 solution was added to each well. Then, the absorbance values ($OD_{450}$) were measured for each well with a microplate reader. The lymphocyte proliferation was interpreted as a stimulation index (SI), which is the ratio of the 450 nm readings between the stimulated wells and the unstimulated wells.

**Cytokine assay.** The measurement of IFN-$\gamma$ and IL-4 production via ELISPOT was performed as previously reported (34). After the preparation of single cell suspensions from the splenic lymphocytes of the mice, ELISPOT assays were performed in 96-well ELISPOT plates. The antibody-coated plates were blocked with complete RPMI 1640 medium for 2 h at room temperature. After blocking, 100 $\mu$L of a lymphocyte suspension ($5 \times 10^5$ cells/well) containing different recombinant proteins (10 $\mu$g/mL) were added to each well. The plates were incubated for 48 h in a humidified incubator at 37°C with 5% $CO_2$. Biotinylated anti-mouse IFN-$\gamma$ and IL-4 detection antibodies (1 $\mu$g/mL, BD Pharmingen Inc., San Diego, CA, USA) were added to the plates and incubated at 37°C for 2 h. The plate-bound secondary antibody was visualized using streptavidin-alkaline phosphatase and the BCIP/NBT substrate (both from Beyotime, China). After drying, the number of spots was analyzed with an Immune Spot Reader (Cellular Technology Ltd.). The results are presented as spot forming units (SFU) per $10^5$ cells.

Next, the transcriptional expression of the IL-4 and IFN-$\gamma$ cytokines were determined via quantitative real-time PCR (qRT-PCR) (60). The total RNA was extracted using the RNA-easy Isolation Reagent (Vazyme) in accordance with the manufacturer's protocol. The synthesis and preparation of the cDNA were conducted using a reverse transcription system (TaKaRa). Then, the cDNA was quantified and analyzed using SYBR Green PCR Master Mix (vazyme). The primers used in the present study are listed in Table 2.

**Body weight and respiratory symptoms.** During the experiment, the body weights of the mice in all of the groups were measured at different time points, and the curves of their body weight changes were drawn. After the challenge, the mice were also monitored for the number of sneezes on a daily basis until 21 days after the challenge. The average number of sneezes per week was calculated and compared among the groups.

**Histopathologic analysis.** Lung specimens were fixed in a 10% formalin solution overnight. The fixed specimens were embedded in paraffin, cut into 3 $\mu$m-thick sections, and then stained with H&E as in a routine histopathological examination. The inflammation of the lungs in the mice and the histological analyses were determined as previously described (2). Mycoplasmal pneumonia lesions were scored (from 0 to 3) based on the lesion size and the severity of the lymphoid tissue hyperplasia. The details of the score points are shown in Table 3.

**Statistical analysis.** The GraphPad Prism version 5.04 software package (Prism) was used for the data analysis. The data are shown as the means $\pm$ standard deviations (SD). The bacterial growth curves, growth halos of the bacteria, the antibody titers, and the cytokine concentrations were analyzed using a one-way analysis of variance (ANOVA). The Tukey multiple comparison test was performed to assess differences between groups. We also used an ANOVA followed by Tukey tests to analyze the weight changes, the numbers of sneezes, and the histopathologic scores. A value of $P < 0.05$ was considered to be indicative of a statistically significant result.

**TABLE 3** Microscopic lung lesion scoring

| Score | Histopathological lesions (0 to 3) |
|---|---|
| 0 | Absence of lesion |
| 1 | The alveolar septum is slightly widened, and there is a small amount of lymphocyte infiltration. The bronchiolar epithelium is regularly arranged, and there is a small amount of inflammatory cell infiltration around the lumen |
| 2 | Light to moderate infiltrates of neutrophils, lymphocytes, and macrophages into the septal wall, bronchus, and alveolar lumen |
| 3 | Presence of perivascular or peribronchiolar lymphoplasmacytic hyperplasia and hyperplasia of edema in the alveoli |

## ACKNOWLEDGMENTS

This work was supported by the National Natural Science Foundation of China (grant numbers 32172802, 31672516, 32002301, 31172300, and 30670079), the State Key Laboratory of Genetically Engineered Veterinary Vaccines (No. AGVSKL-ZD202004), the China Postdoctoral Science Foundation [No. 2019M661953], and a project funded by the Priority Academic Program Development of Jiangsu Higher Education Institutions (PAPD). The funding bodies have not been involved in the design of the study the collection, analysis, or interpretation of the data, or the writing of the manuscript. We thank Roy Curtiss III for kindly providing the $\chi$7213, pRE112, and pYA3493.

No conflicts of interest, financial or otherwise, are declared by the authors.

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
