## [Reviewer comments · Microbiology Spectrum]

Microbiology Spectrum

Oral Immunization with Attenuated *Salmonella Choleraesuis* Expressing the P42, P97 Antigens Protects Mice against *Mycoplasma* *hyopneumoniae* Challenge

Guodong Zhou, Yichen Tian, Jiashuo Tian, Qifeng Ma, Shan Huang, Quan Li, Shifeng Wang, and Huoying Shi

Corresponding Author(s): Huoying Shi, College of Veterinary Medicine, Yangzhou University, Yangzhou 225009, Jiangsu, China

Review Timeline:

Submission Date:	June 23, 2022
Editorial Decision:	September 15, 2022
Revision Received:	October 2, 2022
Accepted:	October 28, 2022

Editor: Mariola Edelmann

Reviewer(s): Disclosure of reviewer identity is with reference to reviewer comments included in decision letter(s). The following individuals involved in review of your submission have agreed to reveal their identity: Qingke Kong (Reviewer #1)

Transaction Report:

DOI: <https://doi.org/10.1128/spectrum.02361-22>

September 15, 2022

Prof. Huoying Shi
College of Veterinary Medicine, Yangzhou University, Yangzhou 225009, Jiangsu, China
12 East Wenhui Ave.
Yangzhou, Jiangsu 225009
China

Re: Spectrum02361-22 (Oral Immunization with Attenuated *Salmonella* Choleraesuis Expressing the P42, P97 Antigens Protects Mice against *Mycoplasma hyopneumoniae* Challenge)

Dear Prof. Huoying Shi:

Link Not Available

Sincerely,

Mariola Edelmann

Journals Department
Reviewer comments:

Reviewer #1 (Comments for the Author):

This manuscript described an attenuated *Salmonella* vaccine delivering *M. hyopneumoniae* pS-P42 or pS-P97, and proved that this candidate live *Salmonella* is a good vaccine for preventing *M. hyopneumoniae* infection, and many experiments performed to prove this point including T, B-cell immune responses, histopathological analysis, etc. This work seems reasonable, and the results seem perfect, but there are several issues to be addressed.

1. In the issue of the proteins expression in *Salmonella* vaccine, if polyclonal antibody were used for detecting these two antigens expression, please show the data, I do not believe that "strains carrying the empty vector plasmids did not produce any protein that reacted with the polyclonal antibody".

2. In figure 3, Antibody response induced by vaccine candidates in mice. The question is why the author used log10 to deal with their data, I think log2 is more reasonable because the titer (double dilution) is adopted for statistical analysis.
3. In figure 4, Cell-mediated immune response and antigen-specific stimulation IFN- γ and IL-4 production. Panel DE seems unnecessary as ELISPOT has already proved that cell-dependent immunity.
4. In figure 6, Histological analysis and pathological scores in the lung. Please show the ruler and magnification, a broad vision for this HE analysis is required to show a similar result.
5. The manuscript said that a live *M. hyopneumoniae* vaccine is available on the market, and the vaccine is not good, why did the author set up a control to compare with the current vaccine? I still believe that a live bacteria vaccine is better than a vectored vaccine delivering only one or two antigens.

Reviewer #2 (Comments for the Author):

M. hyopneumoniae is a very important pathogen for pigs, which causes great loss to pig production world widely. However, currently used vaccines, especially inactivated vaccines, cannot provide satisfactory protection. Therefore, research on new vaccines is very important for this bacterium. This study reported that attenuated *Salmonella Choleraesuis* expressing P42 or P97 antigens protected mice against *M. hyopneumoniae* challenge. The results suggest P42 and P97 proteins as good candidates and the *Salmonella Choleraesuis* as a good vector to deliver antigen as live vector vaccine against *M. hyopneumoniae*.

There are still some questions for the authors:

1. The mouse challenge model is not a common model used in the study of *M. hyopneumoniae*. The author needs to explain why mice were selected. Is there any literature basis? Why the IP and IN inoculation was chosen for challenge? In the discussion part, a paragraph needs to be added to discuss the differences and similarities of the pathogenicity between the mouse model and the pig model, including the possible differences between the vaccine evaluation results on the mouse model and the pig model.
2. Line 422-427: Was P97 full length protein or P97 fragment used in this study? It should be clear here. If it was a segment, specific information should be listed.
3. Did the author evaluate the in vivo growth of the attenuated *Salmonella* vaccines in mice after immunization?
4. The legend of Figure 3 is very confusing now. Which is the detection of anti-P42 antibody or anti-P97 antibody in Fig 3A? Why both pS-P42 and pS-P97 induced same antibody? All the samples should be detected with the same coating protein. The description of "weeks after immunization" in Fig 3A means "weeks after primary immunization" or "weeks after second immunization"? And, it did not clearly point out what is Fig 3D, 3E or 3F in the legend.
5. Line 474: how many animals were used for lymphocyte proliferation response test? The same questions also exist in line 452, how many animals were used for BALF samples collection.
6. Line 514: When was the animal autopsy performed? 21 days after infection? Please be clearly here. The time of autopsy is very critical for a challenge model. Why did the author choose this time point? Have the authors compared the pathological changes at different time points previously?
7. Line 518: How many fields of vision were scored for statistical analysis?
8. The lung lesions induced by *M. hyopneumoniae* varies among different lung lobes or positions. Was there a similar phenomenon in mice? If so, how to avoid these differences when carried out the pathological scoring?
9. Line 521-522. It should be clear what data was analyzed by Student's t test and what data was analyzed by one-way ANOVA.

Staff Comments:

Preparing Revision Guidelines

For complete guidelines on revision requirements, please see the journal Submission and Review Process requirements at <https://journals.asm.org/journal/Spectrum/submission-review-process>. **Submissions of a paper that does not conform to**

Microbiology Spectrum guidelines will delay acceptance of your manuscript. "

Please return the manuscript within 60 days; if you cannot complete the modification within this time period, please contact me. If you do not wish to modify the manuscript and prefer to submit it to another journal, please notify me of your decision immediately so that the manuscript may be formally withdrawn from consideration by Microbiology Spectrum.

In this paper, the authors produced vaccines against *Mycoplasma hyopneumoniae* infection in mice utilizing attenuated *Salmonella Choleraesuis* vectors expressing immunogenic proteins P42 and P97. However, there are other concerns that require attention.

1. It has been reported that attenuated *Salmonella typhimurium* expresses *Mycoplasma pneumoniae* antigens such as p97R1 and NrdF as oral vaccines in mice and pigs (*Microbial Pathogenesis* 2001; 30: 101-110, *Journal of Medical Microbiology* (2006), 55, 923 - 929) and that *Salmonella choleraesuis* expresses *Mycoplasma pneumoniae* antigens (p36, p46, p65 and p97R1-NrdF) in mice (*Acta Microbiologica Sinica*, 01 Sep 2011, 51(9):1270-1277). The novel is limited comparison to previously reported vaccines.
2. In recent years, P97 has been the most investigated adhesion factor. According to reports, *M. hyopneumoniae* adhesion to swine cilia is mediated by the R1 repeat region of the P97 adhesin. The paper does not specify which portion was expressed by the P97 plasmid.
3. *Mycoplasma pneumoniae* is typically transmitted to pigs through the intranasal route. Why did the authors administer the in vivo challenge by intraperitoneal and nasal drop? What is the rationale for intraperitoneal infection selection? What was the infectious dosage for each type? The authors have merely indicated in the procedures that "10⁸ CCU of the F66 strain will be administered intraperitoneally and nasally".
4. Absence of trials evaluating safety, as the author took off the sopB gene to minimize the intestinal inflammatory response of the host induced by the *Salmonella* vector itself.
5. The language must be edited by a native speaker or a professional translation agency. Please correct the inaccuracies such as INF.

Reviewer #1

On behalf of my co-authors, we would like to express our deep thanks to you for the constructive comments and suggestions, which are valuable in improving the quality of our manuscript. We have studied the reviewers' comments carefully and endeavored to revise and improve the manuscript accordingly. We would like to express our deep thanks for your helpful comments and hope that we have now produced a more balance and better account of our work. The main corrections in the paper and the detailed point-by-point responses to the reviewers' comments are listed below. We hope that the corrections will meet with your approval. We have marked the major changes with red in the revised manuscript.

1. Your comments:

In the issue of the proteins expression in Salmonella vaccine, if polyclonal antibody were used for detecting these two antigens expression, please show the data, I do not believe that "strains carrying the empty vector plasmids did not produce any protein that reacted with the polyclonal antibody".

Our response:

Thank you for your valuable suggestions. We are sorry for the inaccuracies in the manuscripts. Based on your explanation, we recognize that the WB description in the last submission is improper. We have revised the manuscript according to your comments (Lane 163-165; Lane 797-799). At the same time, the WB results after polyclonal antibody incubation are also resubmitted, hoping to solve your confusion.

2. Your comments:

In figure 3, Antibody response induced by vaccine candidates in mice. The question is why the author used log₁₀ to deal with their data, I think log₂ is more reasonable because the titer (double dilution) is adopted for statistical analysis.

Our response:

Thank you very much for your constructive comments. This point is very excellent. Based on your comments, we processed and analyzed the antibody titers using log₂. At the same time, Fig. 3 has also been modified, which is believed to be more reasonable.

3. Your comments:

In figure 4, Cell-mediated immune response and antigen-specific stimulation IFN- γ and IL-4 production. Panel DE seems unnecessary as ELISPOT has already proved that cell-dependent immunity.

Our response:

Thanks for your valuable comments. To better demonstrate that the vaccine candidates in this study can induce cell-dependent immune responses, we conducted experiments from multiple perspectives. The ELISPOT experiment was mainly verified from the protein level, and the mRNA level of cytokines was a further verification of this result, which proved the reliability of the result.

4. Your comments:

In figure 6, Histological analysis and pathological scores in the lung. Please show the ruler and magnification, a broad vision for this HE analysis is required to show a similar result.

Our response:

Thank you very much for your constructive comments. The comments were valuable and helpful for revising and improving our paper. Based on your comments, we have supplemented the magnifications and rulers of the pathological sections (Lane 832-833). At the same time, pathological sections with a broad vision are also provided, which are believed to be more helpful to prove the accuracy of our results (Fig. 6).

5. Your comments:

The manuscript said that a live *M. hyopneumoniae* vaccine is available on the market, and the vaccine is not good, why did the author set up a control to compare with the current vaccine? I still believe that a live bacteria vaccine is better than a vectored vaccine delivering only one or two antigens.

Our response:

Thank you for your comment. We fully agree with this view. Indeed, the 168 live vaccine strain (Mhp-168) is one of the Mhp vaccine strains that are used widely in China. And commercial attenuated vaccines based on Mhp-168 have been shown to be effective to decrease respiratory symptoms and increase growth parameters in commercial pig herds. Therefore, the study design would have been more reasonable and the results would have been more powerful if it had set a Mhp-168 control group. In follow-up studies to the current study, we will address this limitation by setting Mhp-168 control group. Like reviewer said, live vaccines are able to provide more protection than vaccines that only provide one or two antigens. Therefore, in follow-up studies, we will screen more antigens. At the same time, it will also be investigated whether the combined use of multiple antigens provides better protection.

Reviewer #2

On behalf of my co-authors, we would like to express our deep thanks to you for the constructive comments and suggestions, which are valuable in improving the quality of our manuscript. We have studied the reviewers' comments carefully and endeavored to revise and improve the manuscript accordingly. We would like to express our deep thanks for your helpful comments and hope that we have now produced a more balance and better account of our work. The main corrections in the paper and the detailed point-by-point responses to the reviewers' comments are listed below. We hope that the corrections will meet with your approval. We have marked the major changes with red in the revised manuscript.

1. Your comments:

The mouse challenge model is not a common model used in the study of M. hypopneumoniae. The author needs to explain why mice were selected. Is there any literature basis? Why the IP and IN inoculation was chosen for challenge? In the discussion part, a paragraph needs to be added to discuss the differences and similarities of the pathogenicity between the mouse model and the pig model, including the possible differences between the vaccine evaluation results on the mouse model and the pig model.

Our response:

Thank you for your comments. Indeed, as you said, mice are not a commonly used model for Mhp research. Conventional pigs have been the only animals widely used in pathogenicity and vaccine evaluation studies of Mhp for a long time (1-3). However, as an animal model, pigs are more expensive and have certain difficulties in operation. Therefore, in this study, we tried to use BALB/c mice with more advantages, such as inexpensive, smaller size, and easy to operate, as an animal model. Moreover, BALB/c mice is the most widely used animal model, and has been used for the pathogenesis and vaccine research of various zoonotic diseases. Moreover, the Jiangsu Academy of Agricultural Sciences in China has successfully established a mouse infection model of Mhp in 2017 and used it to identify the virulence of Mhp (Patent Application No.: CN201711362236.X; Announcement No.: CN108094315A). It is proved that mice can be used as a model of Mhp infection. The choice of the challenge route is because the patent of the Jiangsu Academy of Agricultural Sciences proves that the IP and IN routes can be used to infect mice with Mhp. Thanks for your suggestion, we have added differences in pathogenicity between mouse and pig models to the Discussion section, and also discussed possible differences in vaccine evaluation between the two models (Line 357-382).

2. Your comments:

Line 422-427: Was P97 full length protein or P97 fragment used in this study? It should be clear here. If it was a segment, specific information should be listed.

Our response:

Thank you for your valuable comments. We chose the C-terminal region (R1R2) of

the *P97* gene, which we have supplemented in the manuscript based on your suggestion (Line 400-401).

3. Your comments:

Did the author evaluate the in vivo growth of the attenuated *Salmonella* vaccines in mice after immunization?

Our response:

Thank you very much for your valuable suggestions and comments. In this article, we did not assess the growth of attenuated *Salmonella* vaccine in mice following immunization, but some previous work in our laboratory has done a lot of research on this (4, 5). And it has been demonstrated that the recombinant *Salmonella* Choleraesuis (*S. Choleraesuis*) vector rSC0016 are sufficiently attenuated, and could safely colonize the Peyer's node, and spleen and liver of mice.

4. Your comments:

The legend of Figure 3 is very confusing now. Which is the detection of anti-P42 antibody or anti-P97 antibody in Fig 3A? Why both pS-P42 and pS-P97 induced same antibody? All the samples should be detected with the same coating protein. The description of "weeks after immunization" in Fig 3A means "weeks after primary immunization" or "weeks after second immunization"? And, it did not clearly point out what is Fig 3D, 3E or 3F in the legend.

Our response:

Thanks for your comments, we have revised Fig. 3 and its legend. We tested both P42 and P97 specific antibodies. pS-P42 and pS-P97 are not able to induce the same antibodies, we have made modifications (Fig. 3; Line 805-809). The weeks after immunization in Fig. 3A refer to 2 weeks and 5 weeks after the primary immunization, and we have added corresponding supplements in the manuscript and legend (Line 460-463; Line 808-809). Finally, after modifying Fig. 3, we corresponded the legend and the figure respectively, I believe it can be more convenient for your understanding.

5. Your comments:

Line 474: how many animals were used for lymphocyte proliferation response test? The same questions also exist in line 452, how many animals were used for BALF samples collection.

Our response:

Thank you for your comments and suggestions. Five mice in each group were used for lymphocyte proliferation response and BALF collection, respectively. We have made additions in the manuscript (Line 462-463; Line 484-485).

6. Your comments:

Line 514: When was the animal autopsy performed? 21 days after infection? Please be clearly here. The time of autopsy is very critical for a challenge model. Why did the author choose this time point? Have the authors compared the pathological

changes at different time points previously?

Our response:

Thank you very much for your constructive comments. Yes, we performed the autopsy on 21d post-challenge. This time point was chosen because we detected the levels of inflammatory cytokines and lung pathological changes at different time points after challenge when constructing a mouse challenge model. It was found that the level of inflammatory cytokines produced at 21d was the highest, and the pathological changes of the lungs were the most serious.

7. Your comments:

Line 518: How many fields of vision were scored for statistical analysis?

Our response:

Thank you for your constructive comments. For the scoring of lung inflammation and pathological changes, three fields of vision were selected for each mouse. Finally, the data of five mice in each group were analyzed uniformly.

8. Your comments:

The lung lesions induced by *M. hyopneumoniae* varies among different lung lobes or positions. Was there a similar phenomenon in mice? If so, how to avoid these differences when carried out the pathological scoring?

Our response:

Thanks for the suggestion, this is a great question. In this study, we did not compare whether there were differences in the severity of lesions in different lobes of mice. But we suspect that a similar phenomenon may occur in mice. We will continue to study this issue in the future. If there is a similar phenomenon, we think we can refer to previous studies and score different lung lobes separately (6).

9. Your comments:

Line 521-522. It should be clear what data was analyzed by Student's t test and what data was analyzed by one-way ANOVA.

Our response:

Thanks for your suggestion, we have made changes in the manuscript. The statistical methods used for different data are indicated respectively (Line 532-537).

References

1. Hao F, Bai Y, Xie X, Yuan T, Wei Y, Xiong Q, Gan Y, Zhang L, Zhang Z, Shao G, Feng Z. 2022. Phenotypic characteristics and protective efficacy of an attenuated *Mycoplasma hyopneumoniae* vaccine by aerosol administration. Vaccine doi:10.1016/j.vaccine.2022.08.072.
2. Zong B, Zhu Y, Liu M, Wang X, Chen H, Zhang Y, Tan C. 2022. Characteristics of *Mycoplasma hyopneumoniae* Strain ES-2 Isolated From Chinese Native Black Pig Lungs. Front Vet Sci 9:883416.
3. Djordjevic SP, Eamens GJ, Romalis LF, Nicholls PJ, Taylor V, Chin J. 1997. Serum and mucosal antibody responses and protection in pigs vaccinated against *Mycoplasma*

hyopneumoniae with vaccines containing a denatured membrane antigen pool and adjuvant. Aust Vet J 75:504-11.

4. Li Q, Lv Y, Li YA, Du Y, Guo W, Chu D, Wang X, Wang S, Shi H. 2020. Live attenuated *Salmonella enterica* serovar Choleraesuis vector delivering a conserved surface protein enolase induces high and broad protection against *Streptococcus suis* serotypes 2, 7, and 9 in mice. Vaccine 38:6904-6913.
5. Li YA, Ji Z, Wang X, Wang S, Shi H. 2017. *Salmonella enterica* serovar Choleraesuis vector delivering SaoA antigen confers protection against *Streptococcus suis* serotypes 2 and 7 in mice and pigs. Vet Res 48:89.
6. Villarreal I, Vranckx K, Calus D, Pasmans F, Haesebrouck F, Maes D. 2012. Effect of challenge of pigs previously immunised with inactivated vaccines containing homologous and heterologous *Mycoplasma hyopneumoniae* strains. BMC Vet Res 8:2.

October 28, 2022

Prof. Huoying Shi
College of Veterinary Medicine, Yangzhou University, Yangzhou 225009, Jiangsu, China
12 East Wenhui Ave.
Yangzhou, Jiangsu 225009
China

Re: Spectrum02361-22R1 (Oral Immunization with Attenuated *Salmonella* Choleraesuis Expressing the P42, P97 Antigens Protects Mice against *Mycoplasma hyopneumoniae* Challenge)

Dear Prof. Huoying Shi:

Your manuscript has been accepted, and I am forwarding it to the ASM Journals Department for publication. You will be notified when your proofs are ready to be viewed.

Sincerely,

Mariola Edelmann
Editor, Microbiology Spectrum
